# HkAmsters at CMCL 2022 Shared Task: Predicting Eye-Tracking Data from a Gradient Boosting Framework with Linguistic Features

**Lavinia Salicchi**
The Hong Kong Polytechnic University
lavinia.salicchi@connect.polyu.hk

**Rong Xiang**
The Hong Kong Polytechnic University
xiangrong0302@gmail.com

**Yu-Yin Hsu**
The Hong Kong Polytechnic University
yyhsu@polyu.edu.hk

## Abstract

Eye movement data are used in psycholinguistic studies to infer information regarding cognitive processes during reading. In this paper, we describe our proposed method for the Shared Task of Cognitive Modeling and Computational Linguistics (CMCL) 2022 - Subtask 1, which involves data from multiple datasets on 6 languages. We compared different regression models using features of the target word and its previous word, and target word surprisal as regression features. Our final system, using a gradient boosting regressor, achieved the lowest mean absolute error (MAE), resulting in the best system of the competition.

## 1 Introduction

This year's Cognitive Modeling and Computational Linguistics (CMCL) workshop proposed a Shared Task focused on eye-tracking data prediction (Hollenstein et al., 2022). Differently from the last edition (Hollenstein et al., 2021), the 2022 Shared Task includes two subtasks: "Predict eye-tracking features for sentences of the 6 provided languages" and "Predict eye-tracking features for sentences from a new surprise language". In this paper, we present our proposed method for the first subtask.

In this task, the teams were asked to predict 4 eye-tracking features for 6 different languages (Chinese, Dutch, English, German, Hindi, and Russian); the features were: first fixation duration (FFD, which refers to the duration of the first fixation on the prevailing word), the standard deviation of the FFD across readers, total reading time (TRT, which refers to the sum of all fixation durations on the current word, including regressions), and the standard deviation of TRT across readers.

One of the challenging aspects of this task is the substantially different nature of these languages; they belong to different language families or different branches within the same family (i.e., Germanic, Balto-Slavic, Indo-Iranian,

Sino-Tibetan) and 4 different writing systems are involved (i.e., Latin alphabet, Cyrillic alphabet, Devanagari abugida, and logograms). Therefore, we proposed a unified method that could be applied to account for the similarities and differences exhibited in the datasets of these 6 languages; this method includes regression features of the target word and of its previous word, and the surprisal for the target word within the context. Our codes are shared on Github at: https://github.com/laviniasalicchi/HkAmsters_CMCL2022.

## 2 Related work

Eye movement data provide valuable evidence regarding the cognitive processes underlying reading, and thus revealing how language is elaborated in our brain in every aspect, from morphology (Clifton Jr et al., 2007) to syntax (Van Schijndel and Schuler, 2015) to semantics (Ehrlich and Rayner, 1981). Since the early studies published in the last century, several studies have revealed that some features of the words themselves may influence language processing and, consequently, reading behavior; these features include word position, word length, word frequency, and the number of syllables within the word (Just and Carpenter, 1980). In addition, the *spillover effect* (Rayner et al., 1989) infers that the cognitive load of a word due to its frequency and length (Pollatsek et al., 2008) may influence the processing of its following word. Considering the multilingual nature of this task, in addition to the aforementioned features, we also included whether the word is all in uppercase, and whether it begins with a capital letter.

One additional factor that influences language comprehension is the sentence-level predictability of a word given the previous context (Kliegl et al., 2004), and in recent years, with the growth of computational linguistics, some attempts to model this kind of dynamic have been successfully achieved

using the surprisal (i.e., the negative logarithm of the probability of encountering a word given the context) computed by language models (Hale, 2001; Levy, 2008; Fossum and Levy, 2012).

In the last year's shared task, a regression model was proposed using the following features: two-word features (i.e., word length and word frequency), the cosine similarity between the vector representing the target word and the vector representing the sentence context, and the surprisal computed word-by-word (Salicchi and Lenci, 2021). However, Frank (2017) showed that given the overlaps in the information conveyed by cosine similarity and the surprisal, the latter alone is sufficient for the effective modeling of eye movements. Furthermore, in Frank's model the frequency and length of the word preceding the target one are included in the regression for modeling the spillover effect.

For these reasons, we modified the previously proposed method, increasing the number of word-specific features, and excluding the cosine similarity in our system.

## 3 Datasets

The shared task is formulated as a regression task to predict 2 eye-tracking features and their corresponding standard deviation across readers: (1) FFD; (2) the standard deviation of FFD across readers; (3) TRT; and (4) the standard deviation of TRT across readers. Subtask 1 (multilingual prediction) requires systems to predict these four eye-tracking features of words in 6 provided languages. The dataset includes materials from 8 openly available eye movement corpora:

- **Chinese**: Beijing Sentence Corpus (Pan et al., 2021).

- **Dutch**: GECO Corpus (Cop et al., 2017) .

- **English**: Provo Corpus (Luke and Christianson, 2018), ZuCo 1.0 Corpus (Hollenstein et al., 2018), and ZuCo 2.0 Corpus (Hollenstein et al., 2019).

- **German**: Potsdam Textbook Corpus (Jäger et al., 2021).

- **Hindi**: Postdam-Allahabad Hindi Eyetracking Corpus (Husain et al., 2015).

- **Russian**: Russian Sentence Corpus (Laurinavichyute et al., 2019).

Data statistics are given in Table 1.

| Data Source | Train | Dev | Test |
|---|---|---|---|
| Chinese | 1,355 | 82 | 248 |
| Dutch | 7,462 | 403 | 1,475 |
| English(ZuCo1) | 5,325 | 269 | 994 |
| English(ZuCo2) | 5,398 | 303 | 1,127 |
| English(Provo) | 5,314 | 152 | 440 |
| German | 1,463 | 139 | 293 |
| Hindi | 2,021 | 142 | 433 |
| Russian | 1,140 | 59 | 218 |

Table 1: Dataset statistics. The instance numbers for each portion are given.

## 4 Methodology

In this section, we introduce the selected features, inspired by psycholinguistic studies relying on eye-tracking data, and the investigated regression algorithms. The same set of features was used for each regression model.

### 4.1 Features

Given the multilingual nature of Subtask 1, we adopted several lexical features as hand-crafted features. The **Word position index** was used to provide the sequential information of a word . The **word length** of the current word and previous one was also included. Furthermore, we added two Boolean features: **Capitalization** and **Upper**. The first feature was set to 1 if the first letter of the target word was uppercase, and it was set to 0 otherwise; the second feature was set to 1 if all the letters of the target word were uppercase, and it was set to 0 otherwise. We also used language-specific tools for the following features: **Word frequencies** for all the 6 languages were computed using `wordfreq`[1]. These frequencies were collected for both the current and previous word. **Syllables counts** for Hindi words were computed using the `syllable package`[2] of `Indic NLP Library`, whereas the other languages were available in `textstat`[3]. Finally, to compute **Surprisal**, 6 different GPT versions were used: Russian GPT by Grankin et al.[4], Hindi GPT[5] by Parmar, Chinese GPT[6] by Du (2019), Dutch GPT[7] by

---

[1] pypi.org/project/wordfreq/
[2] github.com/anoopkunchukuttan/indic_nlp_library/find/master
[3] pypi.org/project/textstat/
[4] github.com/mgrankin/ru_transformers
[5] huggingface.co/surajp/gpt2-hindi
[6] github.com/Morizeyao/GPT2-Chinese
[7] github.com/wietsedv/gpt2-recycle

de Vries and Nissim (2021), and German GPT[8]. More specifically, for each word ($w$) we computed the surprisal as the negative logarithm of its probability given the previous context, from the beginning of the sentence to the word immediately preceding the target one:

$$Surprisal(w_n) = -\log(P(w_n|w_0, w_1, ..., w_{n-1}))$$
(1)

with P being the probability computed by GPT.

A total of 9 features were extracted. We decided to generate polynomial features from our set in order to exploit potential interactions. We used the `PolynomialFeatures` functionality of the `scikit-learn` Python package to generate interaction features of order 2, and we used only interaction features, so that the final number of features that were fed to the regressors was 46.

## 4.2 Regressors

Once we had computed all the regression features, we ran several experiments to find the best regression model for each language and each feature, using the mean absolute error (MAE) for all the words within the same language as our main index. We tested several regression algorithms using the implementations in the `scikit-learn` Python package. The adopted scikit-learn API and the main hyper-parameters are listed below:

- **RR** (`Ridge`): Ridge regression solves a regression model in which the loss function is the linear least-squares function, and regularization is given by the l2 norm. `alpha`=1.0, `normalize`=True.

- **MLP** (`MLPRegressor`): The multi-layer perceptron regressor optimizes the squared-loss using L-BFGS algorithm or stochastic gradient descent. `hidden layer size`=5, `activation`=identity, `solver`=adam.

- **PLSR** (`PLSRegression`): PLS regression implements the PLS2, which blocks regression in the case of a one-dimensional response. `components`=5.

- **BRR** (`BayesianRidge`): A Bayesian Ridge model implements the optimization of the regularization parameters lambda and alpha. `alpha_1,alpha_2`==1.0e-6, `lambda_1,lambda_2`=1.0e-6.

[8] huggingface.co/dbmdz/german-gpt2

- **LR** (`LinearRegression`): Linear regression is trained based on an ordinary least-squares function. `normalize`=True.

- **RF** (`RandomForestRegressor`): A random forest is a meta estimator that fits a number of classifying decision trees on various sub-samples of the dataset and uses averaging to improve the predictive accuracy and control over-fitting. `min_samples_split`=2, `min_samples_leaf`=1.

- **SVR** (`SupportVectorRegressor`): SVR is short for epsilon-support vector regression. It uses the kernel trick to map data to map the original data space to a high-dimensional space. `kernel`='rbf', `epsilon`=0.1, `degree`=3.

- **Elast** (`ElastRegressor`): Elast regressor uses linear regression with combined L1 and L2 priors as the regularizer. `alpha`=1.0, `l1_ratio`=0.5, `selection`='cyclic'.

- **LGB** (`LGBMRegressor`): LightGBM is a gradient boosting framework that uses tree-based learning algorithms. It is designed to be distributed and is efficient with faster training speed and higher efficiency. `objective`='regression',`learning_rate`=0.05, `mum_leaves`=31.

## 4.3 Metrics

The performance of the participating systems was evaluated in terms of the mean absolute error ($MAE$), mean squared error ($MSE$), R-Square ($R2$), Pearson correlation ($Pears.$), and Spearman correlation ($Spear.$) between the outputs and the annotated values. In the **Results and Discussion** section, $MAE$ is adopted as the main comparison index.

## 5 Results and Discussion

To use a single model that could be applied to multilingual data, we selected the model with generally better performance. Table 2 shows the performances of different regressors over the FFD for one of the target languages. *LGB*, which is the gradient boosting regressor with the regression feature interacting, provided the best predictions. *LGB* not only had the lowest MAE, but also achieved the best results in terms of MSE, R2, Pearson correlation, and Spearman correlation.

| Regressor | $MAE$ | $MSE$ | $R2$ | $Pears.$ | $Spear.$ |
|---|---|---|---|---|---|
| LGB | 2.31 | 10.35 | 0.20 | 0.48 | 0.45 |
| BRR | 2.44 | 10.77 | 0.17 | 0.42 | 0.37 |
| RR | 2.46 | 10.75 | 0.17 | 0.43 | 0.39 |
| PLSR | 2.47 | 11.17 | 0.14 | 0.38 | 0.33 |
| Elast | 2.51 | 11.52 | 0.11 | 0.34 | 0.32 |
| LR | 2.51 | 11.01 | 0.15 | 0.41 | 0.37 |
| SVR | 2.51 | 11.39 | 0.12 | 0.39 | 0.35 |
| RF | 2.56 | 15.03 | -0.16 | 0.25 | 0.36 |
| MLP | 2.63 | 13.49 | -0.04 | 0.19 | 0.22 |

Table 2: Performance of different regressors over FFD for Hindi. Evaluation metrics including $MAE$, $MSE$, $R2$, $Pears.$, $Spear.$ are provided. LGB is the best performed model for Hindi. BRR and RR are the second and third best models, but the performance gap is rather marginal.

Considering how regressors accounted for each language dataset, we present the lowest MAE values for each feature and each language in Table 3. Despite the generally good performances of LGB1, this model was not always the best. A future direction may be to identify regression features and regression models that are more suitable for a specific language and the relevant eye-tracking features.

This conclusion is reinforced by a further analysis of the performance of our system (Table 6, Appendix); it revealed that TRTAvg was the hardest feature to predict with a mean error across languages of 5.1. Regarding the mean error, the languages that performed better in our model regarding TRTAvg were Dutch (mean error 3.37, standard deviation (std) 3.34) and English (mean error 4.76, std 4.7), but their coefficients of variation were higher, compared with other languages (English: 0.993, Dutch: 0.993), such as Hindi, for which our model registered a high mean error of 8.81 (std 7.045) but the lowest coefficient of variation (0.799). For both Russian and Chinese, LGB1 had high mean errors (approximately 10) and high coefficients of variation (0.867 and 0.931, respectively).

Given the differences in the amount of data among language datasets, our comparison mainly follows the coefficient of variation, which reveals that for FFDAvg, English, German, and Hindi were the languages for which our system performed better, followed by Dutch, Russian and Chinese. For TRTStd, the better performances were on the datasets of Hindi, Chinese, and Russian, whereas the most difficult portions of the dataset for this feature were those in English, Dutch, and German. Finally, regarding the errors for FFDStd, English

| Feature | Language | Model | MAE |
|---|---|---|---|
| FFD | Chinese | ELAST1 | 3.18 |
| | Dutch | LGB0 | 1.72 |
| | English | LGB0 | 5.37 |
| | German | SVR0 | 0.451 |
| | Hindi | LGB1 | 2.31 |
| | Russian | SVR1 | 2.45 |
| FFD sd | Chinese | LGB1 | 3.61 |
| | Dutch | ELAST0 | 1.47 |
| | English | LGB1 | 2.21 |
| | German | SVR1 | 0.45 |
| | Hindi | LGB1 | 2.64 |
| | Russian | SVR1 | 2.43 |
| TRT | Chinese | LR1 | 6.52 |
| | Dutch | LGB0 | 3.34 |
| | English | LGB1 | 8.28 |
| | German | RF0 | 3.052 |
| | Hindi | LGB1 | 5.32 |
| | Russian | SVR0 | 9.71 |
| TRT sd | Chinese | LR1 | 6.84 |
| | Dutch | LGB0 | 2.78 |
| | English | LGB1 | 5.42 |
| | German | RF0 | 2.57 |
| | Hindi | BRR1 | 5.23 |
| | Russian | LGB0 | 6.34 |

Table 3: Best models for each language and feature to be predicted. Models in this table with '0' do not have interaction between regression features while models in this table with '1' take advantage of interaction between regression features.

was undoubtedly the language for which our system performed the best, and it showed the worst results for Chinese and Hindi datasets.

Finally, we performed an ablation study for the German dataset, in order to examine the contribu-

|  | MAE | | MSE | | R2 | | Pearson | | Spearman | |
|---|---|---|---|---|---|---|---|---|---|---|
|  | w/o | w | w/o | w | w/o | w | w/o | w | w/o | w |
| FFD | 0.467- | 0.457 | 0.363- | 0.362 | 0.140- | 0.142 | 0.406- | 0.407 | 0.325- | 0.399 |
| FFD sd | 0.464- | 0.456 | 0.432- | 0.425 | 0.036- | 0.051 | 0.245- | 0.274 | 0.272- | 0.327 |
| TRT | 3.517+ | 3.520 | 29.337+ | 30.181 | 0.628+ | 0.618 | 0.865- | 0.875 | 0.793- | 0.803 |
| TRT sd | 2.892- | 2.872 | 20.397- | 20.090 | 0.510- | 0.517 | 0.780- | 0.788 | 0.717+ | 0.716 |

Table 4: Feature analyses for whether using Capital letters in processing the German dataset. '+' indicates a better performance compared with all features training, while vice verses for '-'.

|  | FFD | FFD sd | TRT | TRT sd |
|---|---|---|---|---|
| Word position index | 0.451+ | 0.463- | 3.520 | 2.981- |
| Word length | 0.452+ | 0.463- | 3.612- | 2.879- |
| Previous word length | 0.456+ | 0.456 | 3.543- | 2.869+ |
| Word log frequency | 0.468- | 0.475- | 3.677- | 3.055- |
| Previous word log frequency | 0.459- | 0.470- | 3.470+ | 2.853+ |
| Uppercase | 0.457 | 0.456 | 3.520 | 2.872 |
| Capitalization | 0.467- | 0.464- | 3.517+ | 2.892- |
| Syllable count | 0.459- | 0.457- | 3.527- | 2.933- |
| Surprisal score | 0.456+ | 0.452+ | 3.573- | 2.889- |
| all | 0.457 | 0.456 | 3.520 | 2.872 |

Table 5: An ablation study for the German dataset (no **Uppercase**). The MAE results are presented using leave-one comparison. '+' indicates a better performance compared with all features training, while vice verses for '-'.

tion of the different features. Table 4 shows the results of whether using the feature **Capitalization**. Despite some minor performance drop (especially for TRT), using **Capitalization** generally improves the evaluation metrics. Table 5 summarizes the MAE results of feature ablation study for the German dataset. In general, every feature incorporated in the proposed system contributes to the best practice. These preliminary results suggest that the features we adopted are tenable. We leave a more comprehensive cross-lingual comparison along this line for the future study.

In summary, in our system, TRTAvg was the most difficult one to predict, but TRTAvg and TRTStd showed better performance in Hindi, and FFDAvg and FFDStd were better in English. Our proposed system outperformed the Shared Task baseline with an average MAE of 3.0112, resulting in the best system of the competition.

## 6 Conclusions

In this paper, we described the system we proposed for the CMCL2022 Shared Task - Subtask 1 on multilingual data. Using a gradient boosting regressor with features of the target words as well as their previous word, and the surprisal between the target word and the previous context as regression features, we predicted two eye-tracking features and two standard deviations: first fixation duration, total reading time, and their standard deviations across readers.

Despite the multilingual nature of this task, we were able to reach our goal of creating a unified system capable of modeling the human reading behavior in 6 substantially different languages. Our results showed a tendency of better performances with FFD related features than with TRT related ones. This may partly reflect the fact that in our system, more word-level hand-crafted features were included, which could favor this token-level prediction task, given that FFD is often assumed to reflect lexical information processing, whereas TRT may be related to a later stage of language processing related to information-structural integration.

## Acknowledgments

We would like to thank the reviewers for their insightful feedback. This research was made possible by the start-up research fund (BD8S) at the Hong Kong Polytechnic University.

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

**Appendix**

|  | FFDAvg | FFDStd | TRTAvg | TRTStd |
|---|---|---|---|---|
| **Chinese** | | | | |
| $Mean$ | 5.884 | 7.152 | 10.096 | 7.804 |
| $Stddev$ | 7.688 | 11.216 | 9.396 | 6.396 |
| $CV$ | 1.307 | 1.568 | 0.931 | 0.820 |
| **Dutch** | | | | |
| $Mean$ | 1.754 | 1.484 | 3.367 | 2.798 |
| $Stddev$ | 1.741 | 1.385 | 3.343 | 2.714 |
| $CV$ | 0.993 | 0.933 | 0.993 | 0.970 |
| **English(Zuco1)** | | | | |
| $Mean$ | 0.960 | 1.010 | 4.180 | 4.102 |
| $Stddev$ | 0.819 | 0.865 | 4.335 | 4.649 |
| $CV$ | 0.853 | 0.856 | 1.037 | 1.133 |
| **English(Zuco2)** | | | | |
| $Mean$ | 1.682 | 1.841 | 4.672 | 4.169 |
| $Stddev$ | 1.383 | 1.459 | 4.805 | 4.169 |
| $CV$ | 0.822 | 0.793 | 1.028 | 1.000 |
| **English(Provo)** | | | | |
| $Mean$ | 2.061 | 2.014 | 5.434 | 5.167 |
| $Stddev$ | 1.682 | 1.857 | 4.969 | 4.872 |
| $CV$ | 0.816 | 0.922 | 0.915 | 0.943 |
| **German** | | | | |
| $Mean$ | 0.457 | 0.456 | 3.520 | 2.872 |
| $Stddev$ | 0.392 | 0.468 | 4.233 | 3.453 |
| $CV$ | 0.857 | 1.025 | 1.203 | 1.202 |
| **Hindi** | | | | |
| $Mean$ | 6.615 | 9.716 | 8.814 | 9.904 |
| $Stddev$ | 6.034 | 11.296 | 7.045 | 7.265 |
| $CV$ | 0.912 | 1.163 | 0.799 | 0.734 |
| **Russian** | | | | |
| $Mean$ | 2.669 | 2.703 | 10.152 | 6.649 |
| $Stddev$ | 2.934 | 1.942 | 8.805 | 6.140 |
| $CV$ | 1.099 | 0.719 | 0.867 | 0.923 |

Table 6: Error analysis on the performance of our proposed system on every portion of the dev dataset. $Mean$ refers to the average of the absolute error of all words in a portion. $Stddev$ refers to the standard deviation of the absolute error of all words in a portion. $CV$ refers to the coefficient of variation (representing a relative standard deviation), which is a statistical measure of the dispersion of the absolute error of all words in a portion.