# OpenReview forum: "HkAmsters at CMCL 2022 Shared Task: Predicting Eye-Tracking Data from a Gradient Boosting Framework with Linguistic Features"
_aclweb.org/ACL/2022/Workshop/CMCL_Shared_Task — CMCL Shared Task_

### Official Review · Reviewer_KK5N · 2022-03-16
**HkAmsters at CMCL 2022 Shared Task: Predicting Eye-Tracking Data from a Gradient Boosting Framework with Linguistic Features**

**Rating:** 7
**Confidence:** 4

**Review:**

The paper describes the authors' regression approach to the first subtask of the CMCL 2022 Shared Task which scored best across all submissions. There are some parts of the analysis that are not clear to me and need, in my opinion, clarification before publication. Particularly how the model has been selected/evaluated. What part of the dataset have you used for that? This is crucial when comparing results from other participants in the shared task.

Further comments/questions:
* binary features for capitalized words were used for all languages, have you evaluated the effect on languages like German where many more words are capitalized than in other languages?
* what is meant with "normalized word index"?
* how many "previous words" are included in the features?

Another comment concerns the number of features, 46 features seems a lot. The model might benefit from some cross-validation for feature selection to prevent overfitting. Have you looked at that?

---

### Official Review · Reviewer_fcRw · 2022-03-25
**A good paper that could use some more explanations**

**Rating:** 7
**Confidence:** 4

**Review:**

The paper describes a regression-based approach for the tasks of CMCL-2022. One of the systems described, achieves the lowest MAE score in the competition. The authors use regression features of a target word and the features of the preceding words along with surprisal of the word for their systems.  A total of 46 features were used as regression features.
The feature description could use some elaboration. For instance, it is not clear what normalized word index is. Also, how surprisal was calculated is not described.
It would have been interesting to have a cross-validation section to check if there was overfitting in the model.

---

### Decision · Program_Chairs · 2022-03-28

Accept